# Positive Correlation between Relative Concentration of Spermine to Spermidine in Whole Blood and Skeletal Muscle Mass Index: A Possible Indicator of Sarcopenia and Prognosis of Hemodialysis Patients

**DOI:** 10.3390/biomedicines11030746

**Published:** 2023-03-01

**Authors:** Hidenori Sanayama, Kiyonori Ito, Susumu Ookawara, Takeshi Uemura, Sojiro Imai, Satoshi Kiryu, Miho Iguchi, Yoshio Sakiyama, Hitoshi Sugawara, Yoshiyuki Morishita, Kaoru Tabei, Kazuei Igarashi, Kuniyasu Soda

**Affiliations:** 1Division of Neurology, First Department of Integrated Medicine, Saitama Medical Center, Jichi Medical University, Saitama 330-8503, Japan; 2Division of Nephrology, First Department of Integrated Medicine, Saitama Medical Center, Jichi Medical University, Saitama 330-8503, Japan; 3Department of Pharmaceutical Sciences, Faculty of Pharmaceutical Sciences, Josai University, Saitama 330-0295, Japan; 4Department of Dialysis, Minami-Uonuma City Hospital, Niigata 949-6680, Japan; 5Division of General Medicine, Department of Comprehensive Medicine 1, Saitama Medical Center, Jichi Medical University, Saitama 330-8503, Japan; 6Amine Pharma Research Institute, Innovation Plaza at Chiba University, Chiba 260-0856, Japan; 7Saitama Medical Center, Jichi Medical University, Saitama 330-8503, Japan; 8Saitama Ken-o Hospital, Okegawa, Saitama 363-0008, Japan

**Keywords:** polyamine, spermine, spermidine, sarcopenia, hemodialysis

## Abstract

Several mechanisms strictly regulate polyamine concentration, and blood polyamines are excreted in urine. This indicates polyamine accumulation in renal dysfunction, and studies have shown increased blood polyamine concentrations in patients with renal failure. Hemodialysis (HD) may compensate for polyamine excretion; however, little is known about polyamine excretion. We measured whole-blood polyamine levels in patients on HD and examined the relationship between polyamine concentrations and indicators associated with health status. Study participants were 59 hemodialysis patients (median age: 70.0 years) at Minami-Uonuma City Hospital and 26 healthy volunteers (median age: 44.5 years). Whole-blood spermidine levels were higher and spermine/spermidine ratio (SPM/SPD) was lower in hemodialysis patients. Hemodialysis showed SPD efflux into the dialysate; however, blood polyamine levels were not altered by hemodialysis and appeared to be minimally excreted. The skeletal muscle mass index (SMI), which was positively correlated with hand grip strength and serum albumin level, was positively correlated with SPM/SPD. Given that sarcopenia and low serum albumin levels are reported risk factors for poor prognosis in HD patients, whole blood SPM/SPD in hemodialysis patients may be a new indicator of the prognosis and health status of HD patients.

## 1. Introduction

The polyamines spermine (SPM), spermidine (SPD), and their precursor putrescine (PUT) are aliphatic amines essential for all living cells (Appendix A). A graph with the chemical formulae of polyamines is presented in Appendix A. Polyamines are synthesized intracellularly from arginine and undergo catabolism by several enzymes. In addition, cells can take up polyamines from their surroundings, and intracellular polyamines are excreted from cells [1]. The involvement of several enzymes in polyamine synthesis and catabolism and the presence of transport systems across the cell membrane suggest that intracellular polyamine concentrations are tightly regulated, which in turn indicates that polyamines are important for cellular homeostasis.

Polyamines have diverse functional profiles and are involved in a variety of biological processes, including chromatin structure remodeling, gene transcription and translation, cell proliferation, and circadian clock regulation. Polyamines act on nucleic acids by interacting with the negatively charged phosphodiester backbone, resulting in conformational changes in DNA and stabilization of tRNA. Polyamines also act as chemical chaperones and may be associated with protein conformational changes and function [2].

Intracellular de novo synthesis is considered the major source of polyamines during the fetal and developmental stages. With growth, the enzymatic activity of polyamine synthase gradually decreases, and the capacity for intracellular polyamine synthesis is reduced. Based on these findings, it can be inferred that polyamine levels decrease with age. Indeed, the titles and abstracts of certain studies describe age-dependent decreases in polyamine concentrations. For example, when the relationship between age and polyamine concentrations is examined at all ages, including in children, blood polyamine levels decline with age [3]. However, this age-dependent decline occurs only in the very early stages of growth; that is, particularly during fetal life. Many studies have clearly shown that polyamine concentrations in adult tissues, blood, and urine do not decline with age [1]. In addition to intracellular de novo polyamine synthesis, cells can take up polyamines from the surrounding environment. The most important source of polyamines in adults is the gastrointestinal tract; that is, polyamines in ingested food are synthesized in the gastrointestinal tract through the action of intestinal bacteria. In recent years, changes in blood SPM concentrations have been observed with the continued consumption of a high-polyamine diet in healthy volunteers [4]. 

However, polyamines in the body are excreted in urine via the kidneys. Polyamine synthesis is enhanced by the autonomous activation of polyamine synthase enzymes in cancer cells, and polyamines synthesized by cancer cells are transferred to blood cells [1]. It is well known that increased polyamine excretion is observed in cancer patients [5]. This also means that polyamines in the body are not eliminated via the kidneys in patients with renal failure, and high blood polyamine levels have been observed in patients with renal failure [6]. Hemodialysis (HD) is used to treat uremia in patients with renal failure. However, little is known about polyamine efflux into the dialysate or the changes in blood concentrations caused by HD. One purpose of this study was to compare blood polyamine levels between patients with chronic renal failure (CKD) undergoing HD and healthy volunteers. The effects of HD on blood polyamine levels were examined before and after HD. 

Several researchers have been reported to correlate with degree of aging and may be predictive of prognosis in HD patients as well as in otherwise healthy individuals [7,8]. Recent studies have shown that blood polyamine levels and the SPM-to-SPD ratio are associated with health status and various age-related chronic diseases [4,9]. We examined the relationship between polyamine concentration and indicators reported to be associated with healthy life expectancy. Additionally, this study examined whether blood polyamine levels are an indicator of both health status and degree of senescence in dialysis patients.

## 2. Materials and Methods

### 2.1. Study Design and Participants

The study was conducted in accordance with the Declaration of Helsinki (revised in Tokyo in 2004) and was approved by the Institutional Review Board of Minami-Uonuma City Hospital (ID: R1-13). Both adult patients undergoing hemodialysis for renal failure and healthy adult volunteer participants were fully informed about the study and provided informed consent before participating. All patients received intermittent hemodialysis (HD) or hemodiafiltration (HDF) three times per week for 3–5 h. Participants on HD or HDF with pacemaker implants were excluded from the study because their skeletal muscle mass could not be measured using bioelectrical impedance analysis (BIA). Healthy volunteers were adults over 20 years of age with no specific illnesses who were recruited through notices such as posters. Healthy volunteers underwent annual medical checks (laboratory tests: blood count, blood glucose, HbA1c, creatinine, and blood urea nitrogen levels) and had no history of regular hospital visits or oral medications. Because the presence of neoplasms has a significant impact on polyamine levels, patients with cancer and those with a history of cancer treatment within 3 months were excluded from the study.

### 2.2. Data and Sample Collection

Medical backgrounds and other relevant data of the HD and HDF patients were collected from their medical records, such as information on the causative diseases of renal failure and dialysis, including dry weight (DW). The health status of the healthy adult volunteers was determined through interviews and physical examinations.

Body mass index (BMI) was calculated as body weight in kilograms divided by height in meters squared. Multifrequency bioelectrical impedance analysis (MF-BIA) was performed using InBody S10 (InBody Japan, Tokyo, Japan) to measure body composition and muscle mass. Skeletal muscle mass index (SMI) was calculated by dividing the limb skeletal muscle mass in kilograms by height in meters squared. Hand grip strength (HGS) was measured using a hand grip strength meter (Smedley’s Hand Dynamo Meter, Matsumiya Ika Seiki, Tokyo, Japan) in a sitting position; the highest numerical value measured in the left and right hand grip strength tests was used.

Blood samples were obtained at ambient temperature from the patients before and after HD or HDF. For the measurement of polyamine concentrations in the dialysate from dialysis patients, the dialysate was collected from six dialysis patients before dialysis began, 30 min after dialysis began, and just before the end of dialysis. For both HD patients and volunteers, whole blood samples for the measurement of polyamine concentrations (as well as dialysate collected from dialysis patients for the polyamine assay) were stored at −20 °C until the assay.

### 2.3. Blood and Biochemical Tests

All blood sample measurements, except those for polyamines, were performed in the clinical laboratory of Minami-Uonuma City Hospital. Hemoglobin (Hb) concentration was measured using the sodium lauryl sulfate hemoglobin detection method with an automated blood cell counter (XN-2000; Sysmex Corporation, Kobe, Japan). Biochemical blood tests were performed using an automated biochemical analyzer (BM6070, JEOL Ltd., Akishima, Japan). Serum albumin (Alb) concentration was measured using the modified bromocresol purple (BCP) method. Blood urea nitrogen (BUN) and creatinine (Cre) levels were measured using enzymatic methods. C-reactive protein (CRP) levels were measured using the latex immunoturbidimetric method. Serum calcium (Ca) and phosphorus (P) levels were measured using Arsenazo III colorimetric and enzymatic calorimetric methods, respectively. 

### 2.4. Determination of Polyamine Concentrations in Whole Blood 

Since most polyamines in the body are associated with cells, and the majority of blood polyamines are also attached to blood cells [10], we measured polyamine concentrations in whole blood. Whole blood samples were obtained from the arteriovenous fistula before and after HD., and then heparinized, collected, and stored at −20 °C. For polyamine measurement, whole blood was thawed and degraded by sonication and freeze-thaw cycles. Polyamine concentrations were determined using high-performance liquid chromatography (HPLC) at the Cardiovascular Institute for Medical Research at Saitama Medical Center of Jichi Medical University [11]. To extract polyamines, whole blood was diluted five times with 5% trichloroacetic acid (TCA). Subsequently, adjusted samples were incubated at 95 °C for 45 min. After centrifugation at 13,000× *g* for 20 min at 4 °C, the supernatant was collected and deproteinated by increasing the TCA concentration to 10%. Next, incubation was carried out at 95 °C for 45 min, followed by centrifugation at 13,000× *g* for 20 min at 4 °C. The polyamines in 20 µL of TCA supernatant were separated with an HPLC system (Shimadzu Corporation, Kyoto, Japan) with a TSKgel Polyaminepak column (column size 4.6 mm ID × 50 mm length, particle size 7 µm, TOSOH Bioscience, Tokyo, Japan) at 50 °C. The flow rate of the separation buffer (0.09 M citric acid (Nacalai Tesque, Inc., Kyoto, Japan), 2 M NaCl (Nacalai Tesque, Inc.), 0.64 mM n-capric acid (Nacalai Tesque, Inc.), 0.1% Brij-35 (Sigma-Aldrich Japan, Tokyo, Japan), 20% methanol (FUJIFILM Corporation, Osaka, Japan), adjusted to pH 5.10) was 0.42 mL/min. Polyamines were detected by fluorescence intensity after the reaction of the column effluent at 50 °C with a solution containing 0.4 M boric buffer (pH 10.4) (Nacalai Tesque, Inc.), 0.1% Brij-35, 2.0 mL/L 2-mercaptoethanol (Nacalai Tesque, Inc.), and 0.06% o-phthalaldehyde (Nacalai Tesque, Inc.). The flow rate of the o-phthalaldehyde solution was 0.42 mL/min, and fluorescence was measured at an excitation wavelength of 340 nm and an emission wavelength of 455 nm. The retention times for SPD and SPM were 12 min and 23 min, respectively. The concentrations in the original whole blood samples were expressed as micromolar concentrations (µM).

### 2.5. Determination of Polyamine Concentrations in Dialysate

Dialysates were collected from the waste stream before dialysis started, 30 min after dialysis started, and immediately before the end of dialysis. Immediately after collection, the dialysate was frozen at −20 °C and thawed before measurement. For the extraction of polyamines, dialysate containing 5% TCA was incubated at 95 °C for 45 min. The HPLC assay was performed under the same conditions as those used for the blood samples. In a preliminary study, the detection limits of SPD and SPM concentrations in the dialysate were 0.5 nM and 5 nM, respectively. The flow rate of the dialysate was 500 mL/min. The total volume of the dialysate was determined by the time required for dialysis, and the amount of polyamine excreted in the dialysate was calculated using the polyamine concentration measured by HPLC.

### 2.6. Statistical Analysis

The Shapiro–Wilk normality test was used to determine the normal distribution of the dataset. Continuous variables with normal distributions were expressed as mean ± standard deviation, and non-normal variables were expressed as median and interquartile ranges. A paired Student’s *t*-test was used to compare normally distributed values. For non-normally distributed values, the Wilcoxon signed-rank test was used to compare the matched polyamine concentration data before and after dialysis. Additionally, the Mann–Whitney U test was used to compare non-matched data between dialysis patients and healthy controls.

Correlations between the two groups were evaluated using Pearson’s correlation for normally distributed data and Spearman’s rank correlation for non-normally distributed data. All analyses were performed using IBM SPSS Statistics for Windows, version 28.0 (IBM Corp., Armonk, NY, USA). A *p* value < 0.05 was considered statistically significant.

## 3. Results

### 3.1. Characteristics of the Participants

A flowchart of the patient registration process is shown in Appendix A. Overall, 11 patients refused to participate in the study, two patients died before the end of the study, one patient did not meet the requirements for participation because he was wearing a pacemaker, and six patients who had a past or current history of cancer were excluded. A total of 59 patients (35 men and 24 women) were enrolled in the study (Appendix A).

The backgrounds and characteristics of the participants are presented in Table 1. Many previous research results have shown that polyamine concentrations do not change with age in healthy adults [1]. Therefore, we did not focus on matching the age range of the control group with that of the target group, but respected the volunteers’ willingness to participate; as a result, the control and dialysis groups were not matched for age. Their median age was 70.0 years (range, 62–75 years), and their median duration from the introduction of hemodialysis was 62 months (range, 29–142 months). The causative diseases requiring dialysis were glomerulonephritis (*n* = 20), diabetes mellitus (*n* = 18), nephrosclerosis (*n* = 6), and other diseases (*n* = 15). Forty-four patients received HD, and 15 patients received HDF. Twenty-six healthy volunteers (11 men and 15 women) participated in the study as controls, and none were excluded from the study. Their median age was 44.5 years (range: 37–52 years). There was no difference in sex (*p* = 0.149) between the dialysis and control groups, whereas dialysis patients were significantly older in (*p* < 0.001). HD patients had significantly lower SMI (6.3 ± 1.1 vs. 8.0 ± 1.2, *p* < 0.001) and HGS (23.5 ± 9.0 vs. 35.3 ± 9.6, *p* < 0.001) values than healthy individuals (Table 1).

### 3.2. Comparison of Blood Polyamine Levels

The blood SPD concentrations in HD patients were higher than those in healthy participants (10.1 (6.9–12.7) µM vs. 6.0 (4.8–7.1) µM, *p* < 0.001). The numerical values of blood SPM concentrations in HD patients were higher than those in healthy volunteers; however, there was no significant difference between them (4.4 (2.6–5.8) µM vs. 3.6 (2.7–4.9) µM, *p* = 0.401). The blood SPM/SPD ratio in HD patients was significantly lower than that in healthy participants (0.43 (0.34–0.59) vs. 0.64 (0.48–0.79) µM; *p* < 0.001) (Figure 1).

SPD and SPM concentrations and SPM/SPD ratios were evaluated before and after HD (Figure 2). HD did not significantly affect the SPD or SPM levels in whole blood; however, the SPM/SPD ratios decreased after HD (0.43 (0.34–0.59) vs. 0.42 (0.32–0.54), *p* < 0.001).

### 3.3. Polyamine Concentrations in Dialysate

HPLC did not detect (0.5 and 5 nM below the detection limit for SPD and SPM, respectively) either SPD or SPM in the dialysate before dialysis. SPD concentrations were detected in the dialysate 30 min after the start of dialysis for all six patients; the mean SPD concentration in the dialysate was 0.21 ± 0.13 μM. However, only four of the six patients had detectable SPD concentrations in the dialysate immediately before the end of dialysis. The mean concentration of the four samples was 0.14 ± 0.13 μM. SPM was not detected in any of the samples.

### 3.4. Analyses of Various Measurements in HD Patients 

Various correlations between clinical parameters (excluding polyamines) were examined (Table 2). Patient age was negatively associated with SMI (ρ = −0.493, *p <* 0.001), HGS (ρ = −0.422, *p =* 0.001), Alb (ρ = −0.276, *p =* 0.034), and Cre (ρ = −0.427, *p =* 0.001). HD duration was negatively correlated with BMI (ρ = −0.420, *p* = 0.001) and HGS (ρ = −0.293, *p* = 0.024), but positively correlated with Cre (ρ = 0.325, *p* = 0.012). BMI and SMI were positively correlated (r = 0.377, *p = 0.003*). SMI was positively correlated with HGS (r = 0.629, *p* < 0.001), Alb (ρ = 0.281, *p =* 0.031), and Cre (ρ = 0.350, *p =* 0.007). HGS was positively correlated with Alb (ρ = 0.372, *p =* 0.004) and Cre (ρ = 0.405, *p* = 0.001). The Hb and CRP levels were negatively correlated (ρ = −0.284, *p =* 0.029). Alb was negatively correlated with CRP (ρ = −0.480, *p <* 0.001) and positively correlated with Ca (ρ = 0.463, *p <* 0.001). BUN was positively correlated with Cre (ρ = 0.354, *p =* 0.006) and *p* levels (ρ = 0.390, *p =* 0.002).

### 3.5. The Relationship between Polyamines and Various Measurements

The SPD and SPM concentrations and SPM/SPD ratios in relation to other clinical parameters are shown in Table 3. Notably, age and BMI did not correlate with either SPD and SPM concentrations or the SPM/SPD ratio. No significant correlation was found between HD duration and SPD, SPM, or the SPM/SPD ratio. The SPM/SPD ratio was significantly and positively correlated with SMI (ρ = 0.309, *p* = 0.017) and HGS (ρ = 0.260, *p =* 0.046) in a simple linear regression analysis (Figure 3). SMI did not correlate with SPM or SPD, and HGS showed no significant association with SPM or SPD.

## 4. Discussion

In this study, we determined that whole-blood SPD levels were higher in HD patients than in healthy participants; however, SPM levels did not differ between groups. The results of the present study are similar to those reported previously. Several reports have shown that erythrocyte SPD concentrations are significantly higher in patients with chronic renal failure than in healthy participants; however, their SPM levels are comparable to those in healthy participants [6]. Moulinoux et al. reported a significant increase in erythrocyte SPD levels in patients undergoing hemodialysis, whereas SPM levels were abnormally high in only a small proportion of patients [12]. Decreased urinary polyamine excretion in both patients with chronic renal failure and those on HD is considered one of the major causes of polyamine accumulation in blood cells.

Polyamine excretion from the dialysate was confirmed. Previous reports have shown that serum polyamine levels decrease after dialysis [13], indicating that HD can lead to the excretion of polyamines. However, in the present study an increase in polyamine excretion in the dialysate was mainly observed immediately after the start of dialysis, without affecting polyamine concentrations in whole blood. This suggests that polyamine excretion into the dialysate was insufficient to reduce whole-blood polyamine levels, as previously reported [6]. Given that most blood polyamines are present in association with blood cells, this may be due to the loss of intracellular polyamine delivery from blood cells to renal cells through intercellular contact. 

Because of the noticeably higher SPD, the SPM/SPD ratio in HD patients was significantly lower than that in healthy volunteers. A lower SPM/SPD ratio in HD patients has also been previously reported [6,14]. In this study, HD patients were significantly older on average than healthy volunteers. Several reports have shown that the SPM/SPD ratio tends to decrease in healthy individuals because of the absence of an age-related decrease in SPD concentration and the presence of an age-related decreasing trend in SPM concentration [4,9,15]. However, these changes were minor and not statistically significant. Therefore, although the age difference between the two groups in this study may have had some effect on both polyamine SPM concentrations and SPM/SPD ratios, this is not expected to have a significant effect on the SPM/SPD ratio, especially the SPD concentration.

The findings of this study are consistent with those of previous studies. The difference in age between dialysis patients and controls resulted in a significant difference in muscle mass, as indicated by SMI, and muscle power, as indicated by HGS. This age-related change was also observed in a study involving patients on dialysis alone. In other words, SMI and HGS were negatively correlated with age. Furthermore, Alb and Hb levels decreased with age. The finding that CRP levels are negatively correlated with Alb and Hb levels is consistent with the finding that inflammation inhibits Alb and Hb synthesis. SMI and HGS, which were strongly correlated, were also positively correlated with Alb, a representative indicator of nutritional status, and Cre, the values of which are affected by muscle mass. Previous reports have indicated that several factors correlate with the prognosis of dialysis patients [16,17,18]. Among these, sarcopenia (for which the major diagnostic criteria are gait speed, HGS, and muscle mass [19,20]) is a major factor associated with a poor prognosis [21,22].

The intracellular concentrations of polyamines are tightly regulated by their import, export, synthesis, and catabolism. As part of this process, SPD and SPM are mutually converted [1], and because of this the SPM/SPD ratio has the potential to change. Chronic inflammation is known to be closely related to the development of various age- and lifestyle-related diseases [23,24]. It has also been recognized as the main component of the uremic phenotype linked to cardiovascular diseases and protein-energy wasting, which leads to sarcopenia [25,26] and is a strong predictor of poor outcomes in dialysis patients [27]. Inflammation activates spermidine/spermine-*N^1^*-acetyltransferase (SSAT), which acetylates the primary N1 amines of SPD and SPM. Both acetylated SPM and acetylated SPD are subsequently oxidized by *N^1^*-acetylpolyamine oxidase to produce SPD or putrescine, depending on the starting substrate, with H_2_O_2_ and aldehyde 3-acetoaminopropanal as byproducts [28,29,30]. Inflammation also activates the alternative enzyme spermine oxidase (SMO), which directly converts SPM to SPD while producing H_2_O_2_ and aldehyde 3-aminopropanal [31]. The generated 3-aminopropanal is spontaneously deaminated to produce acrolein, a highly toxic aldehyde [32]. Increased plasma polyamine oxidase activity and elevated acrolein levels have been reported in patients with chronic renal failure [12,33]. Briefly, two molecules of SPM become two molecules of SPD through two enzymes: SSAT and SMO. Simultaneously, one molecule of SPD becomes one molecule of putrescine via SSAT because it does not act on SPD. The inflammation-induced catabolism of SPM was accelerated more than that of SPD, resulting in a decrease in the SPM/SPD ratio (Appendix A). The mechanism of both the increase in SPD and decrease in SPM/SPD in hemodialysis patients needs further investigation; however, similar findings have been reported in neurodegenerative diseases in which chronic inflammation is present as a pathological background. Decreased SPM/SPD ratios have been reported in patients with neurodegenerative diseases and cerebral ischemia associated with elevated SPD [9,34,35].

Recently, the relationship between muscles and polyamines has been examined in various ways. SPD and SPM were significantly decreased in the skeletal muscle of aged mice compared with that of young mice [36]. In contrast, a study of changes in skeletal muscle polyamine concentrations during exercise in male rats showed exercise-induced endogenous testosterone, followed by polyamine synthesis [37]. Polyamines are thought to be involved in skeletal muscle atrophy and hypertrophy and are of great interest in the prevention and treatment of muscle diseases [38,39]. Interventions for polyamine metabolism have the potential to be therapeutic interventions for muscle atrophy; however, further studies are needed. 

Novel findings of this study include the positive correlations between both the SPM/SPD ratio and SMI and between the SPM/SPD ratio and HGS in HD patients. As SMI and HGS (which are also indicators of sarcopenia) are used to predict the prognosis of patients with HD, the SPM/SPD ratio may be a new predictor of prognosis and health status in patients with HD.

## Figures and Tables

**Figure 1 biomedicines-11-00746-f001:**
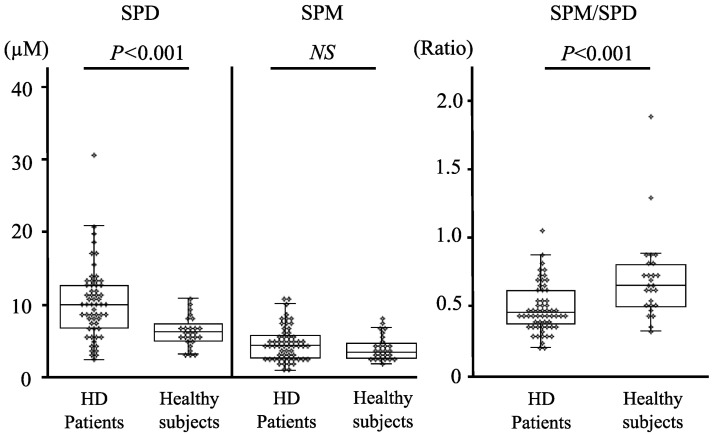
A box and whisker plot and scatterplot of whole blood SPD, SPM, and SPM/SPD in dialysis patients and healthy participants. For each box, the interior line shows the median, and the edges of the box are the estimates of the first and third quartiles. The whiskers extend to the most extreme data points not considered outliers (1.5 times the interquartile range from the edges of the box). Black dots indicate the values for each patient. Left: Comparison of whole-blood SPD and SPM levels in dialysis patients and healthy participants. Right: Comparison of whole blood SPM/SPD between dialysis patients and healthy participants. SPD: Spermidine, SPM: Spermine, HD: hemodialysis, NS: not significant.

**Figure 2 biomedicines-11-00746-f002:**
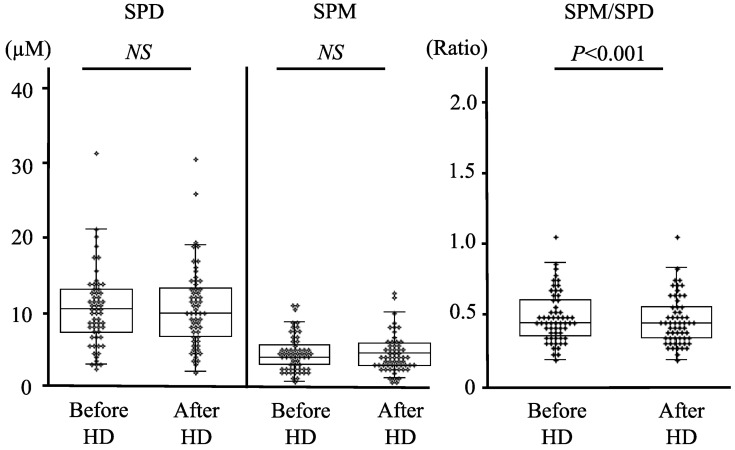
Box and whisker plot and scatter plot of whole-blood SPD, SPM, and SPM/SPD before and after hemodialysis. For each box, the interior line shows the median, and the edges of the box are the estimates of the first and third quartiles. The whiskers extend to the most extreme data points not considered outliers (1.5 times the interquartile range from the edges of the box). Black dots indicate the values for each patient. Left: SPD and SPM in whole blood before and after HD. Right: SPM/SPD in whole blood before and after HD. SPD: Spermidine, SPM: Spermine, HD: hemodialysis, NS: not significant.

**Figure 3 biomedicines-11-00746-f003:**
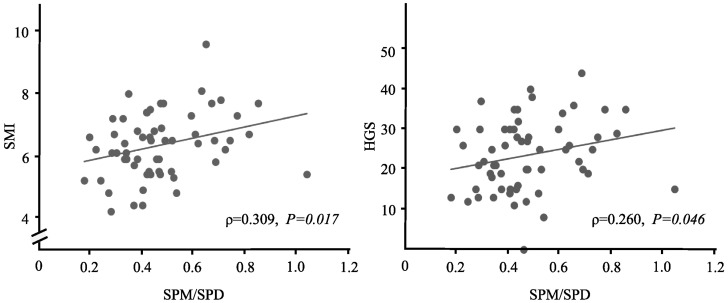
SPM/SPD ratio positively correlated with the SMI and HGS. There was a significant positive correlation between the pre-dialysis whole-blood SPM/SPD ratio and SMI (**left**) and between the SPM/SPD ratio and HGS (**right**) in dialysis patients. The correlation coefficients (ρ) were calculated using Spearman’s rank correlation coefficient. SPD: Spermidine, SPM: Spermine, SMI: Skeletal muscle mass index.

**Table 1 biomedicines-11-00746-t001:** Characteristics of study participants.

Characteristics	HD Patients	Controls
Number of patients *n*, (men/women)	59 (35/24)	26 (11/15)
Age (years)	70.0 (62.0–75.0)	44.5 (37.0–52.0)
HD duration (months)	62.0 (29.0–142.0)	
Dialysis time (min)	246 (246–247)	
Dry Weight (kg), mean ± SD	57.2 ± 11.4	
Interdialytic weight gain (kg), mean ± SD	2.6 ± 1.0	
Interdialytic weight gain/dry weight (%), mean ± SD	4.6 ± 1.7	
BMI (kg/m^2^), mean ± SD	22.4 ± 3.7	22.8 ± 3.1
SMI (kg/m^2^), mean ± SD	6.3 ± 1.1	8.0 ± 1.2
HGS (kg), mean ± SD	23.5 ± 9.0	35.3 ± 9.6
Primary Disease, *n* (%)		
Chronic glomerulonephritis	20 (34)	
Diabetes mellitus	18 (31)	
Nephrosclerosis	6 (10)	
Others	15 (25)	
Past Medical History, *n* (%)		
Diabetes mellitus	22 (37)	0 (0)
Dyslipidemia	27 (44)	0 (0)
Hypertension	55 (93)	0 (0)
Cardiovascular disease	8 (14)	0 (0)
Cerebrovascular disease	12 (20)	0 (0)

Values are shown as the median (interquartile range) unless otherwise indicated. HD, hemodialysis; SD, standard deviation; BMI, body mass index (kg/m^2^); SMI, skeletal muscle mass index (kg/m^2^); HGS, handgrip strength (kg).

**Table 2 biomedicines-11-00746-t002:** Correlations between clinical parameters (excluding polyamine concentrations).

	Age	HD Duration	BMI	SMI	HGS	Hb	Alb	BUN	Cre	CRP	Ca	P
Age	1.000											
HD duration	−0.099	1.000										
BMI	−0.247	−0.420 *	1.000									
SMI	−0.493 **	−0.249 *	0.377 ^†^*	1.000								
HGS	−0.422 *	−0.293 *	0.203 ^†^	0.629 ^†^**	1.000							
Hb	−0.083	0.158	−0.252 ^†^	0.022 ^†^	0.086 ^†^	1.000						
Alb	−0.276 *	−0.049	0.213	0.281 *	0.372 *	0.143	1.000					
BUN	−0.210	−0.068	0.092	0.153	0.237	0.192	0.136	1.000				
Cre	−0.427 *	0.325 *	0.051	0.350 *	0.405 *	0.164	0.210	0.354 *	1.000			
CRP	0.170	0.132	−0.083	−0.156	−0.184	−0.284 *	−0.480 **	−0.038	−0.163	1.000		
Ca	−0.067	0.184	−0.052 ^†^	−0.036 ^†^	−0.024 ^†^	0.229 ^†^	0.463 **	0.135	0.051	−0.077	1.000	
P	−0.132	−0.018	0.183	0.119	0.107	−0.032	−0.026	0.390 *	0.186	0.066	0.039	1.000

^†^ Pearson’s correlation coefficient. ** Correlation is significant at the 0.01 level (2-tailed). * Correlation is significant at the 0.05 level (2-tailed). HD, hemodialysis; BMI, body mass index (kg/m^2^); SMI, skeletal muscle mass index (kg/m^2^); HGS, hand grip strength (kg); Hb, hemoglobin (g/dL); Alb, albumin (g/dL); BUN, blood urea nitrogen (mg/dL); Cre, creatinine (mg/dL); CRP, C-reactive protein (mg/dL); Ca, calcium (mg/dL); P, phosphorus (mg/dL).

**Table 3 biomedicines-11-00746-t003:** Correlations between clinical parameters and polyamine.

	vs. SPD Values in Simple Linear Regression		vs. SPM Values in Simple Linear Regression		vs. SPM/SPD Values in Simple Linear Regression	
	ρ	*p* value	ρ	*p* value	ρ	*p* value
Age	−0.055	0.680	−0.137	0.301	−0.145	0.275
HD duration	−0.032	0.807	−0.106	0.424	−0.170	0.198
BMI	0.113	0.395	0.244	0.063	0.172	0.194
SMI	−0.068	0.610	0.084	0.528	0.309	0.017 *
HGS	−0.159	0.230	−0.063	0.636	0.260	0.046 *
Laboratory findings						
Hb	−0.168	0.203	−0.075	0.570	0.187	0.157
Alb	−0.238	0.070	−0.215	0.102	0.068	0.610
BUN	0.019	0.885	0.044	0.743	0.046	0.727
Cre	0.024	0.857	0.126	0.340	0.115	0.385
CRP	0.219	0.095	0.124	0.351	−0.058	0.663
Ca	−0.072	0.588	−0.068	0.607	0.034	0.799
P	0.046	0.730	0.136	0.303	0.095	0.473

* Correlation is significant at the 0.05 level (two-tailed). As the values were not normally distributed, Spearman’s rank correlation was used to test correlations with polyamine concentration. HD, hemodialysis; BMI, body mass index (kg/m^2^); SMI, skeletal muscle mass index (kg/m^2^); HGS, hand grip strength (kg); Hb, hemoglobin (g/dL); Alb, albumin (g/dL); BUN, blood urea nitrogen (mg/dL); Cre, creatinine (mg/dL); CRP, C-reactive protein (mg/dL); Ca, calcium (mg/dL); P, phosphorus (mg/dL); SPD, spermidine (μM); SPM, spermine (μM).

## Data Availability

The data that support the findings of this study are available from the corresponding author (Kuniyasu Soda), upon reasonable request.

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
