# Peer review of "Positive Correlation between Relative Concentration of Spermine to Spermidine in Whole Blood and Skeletal Muscle Mass Index: A Possible Indicator of Sarcopenia and Prognosis of Hemodialysis Patients"

_biomedicines, 2023, doi:10.3390/biomedicines11030746_

Round 1

Reviewer 1 Report

The study reported in the manuscript "Correlation between relative concentration of spermine to spermidine in whole blood and skeletal muscle mass index: a possible indicator of sarcopenia and prognosis of hemodialysis patients" by Sanayama and colleagues investigated the excretion of polyamines during hemodialysis and the correlation of SPM/SPD ratio with physiological indicators of health status, in particular those associated with sarcopenia. The study concludes that (1) some SPD is excreted into the dialysate, and (2) among the hemodialysis patients, SPM/SPD ratio was positively correlated with hand-grip strength and skeletal muscle index. These results suggest that SPM/SPD ratio may have prognostic value in this patient population. 

However, results that form the basis of the study are flawed and should be improved if the manuscript is to be accepted. Most importantly, there is a very significant difference in median age between the healthy volunteer group and the hemodialysis patients. As all of the indicators measured between the groups can vary with age (polyamine levels, hand-grip strength, SMI), this fact makes the data regarding changes in these factors in hemodialysis patients uninterpretable (Figure 1 and S1, Table S1). As the authors note that the changes in polyamines have been previously reported in HD patients, in my opinion, the data presented here detract merit from  the more solid aspects of the study.

lines 48 - 51: regarding decreased polyamine synthesis and increased uptake, references are needed and more specifics would be helpful, i.e., how do polyamine levels change with age in the blood vs urine vs tissue?

lines 58-59: polyamines synthesized by cancer cells are transferred to blood cells needs a reference

Figure 2: The true significance of this change is ratio is questionable considering the minimal extent of the change as well as the fact that the median values of SPD and SPM are changing in such a manner that one would expect the ratio to increase after dialysis, which is further supported by the presence of SPD in the dialysate. Consider rewording lines 30-31 in the abstract starting with "hemodialysis..." to more accurately convey how minute this change was. 

lines 257-258: needs rewording, as written, it indicates no correlation between SMI and HGS

lines 265-266 remove lipids not in the table from the figure legend.

line 331: sentence regarding mechanism is incomplete

Discussion: it's interesting there is no association between age and polyamine levels within the HD groups. Is this due to the pathology? Might re-analyzing your data based on causal pathology provide more insight into the changes or lack thereof observed before/after HD, etc?

Reviewer 2 Report

In their submission the authors examine the relationship between levels of the polyamides spermine and spermidine in whole blood, physiologic markers of "health", and clearance via hemodialysis. As indicated by the authors in the discussion section the results are principally in keeping with the published literature making the study not overtly novel. However, this topic is of high interest to the special issue.

Major points:

1) The control subjects are not age matched and therefore not appropriate controls. This should, at a minimum, be commented on in the results section, esp. given how many markers of "health" are known to be age related. Similarly, this could be commented on in the discussion, for example Kirkwood's blood panel for frailty vs. what was studied here.

2) Subject characteristics (Sup Table 1) needs to be moved into the main text/display items so readers can see the strengths and limitations of the study design.

3) Significant P values in Table 2 need to be noted as they are in Table 1. On first read I did not easily see anything as being significant, which is not consistent with your findings and text.

Minor points:

1) The introduction and/or discussion could include some information about the past, established role of polyamides as "chaperones" and/or "charge stabilizers" e.g. helping maintain membranes and buffer DNA charge issues.

2) Could comment on SPD to SPM conversion as potentially a cause of ratio changes in discussion as not all readers will know the pathway.

3) Could link the altered polyamine levels and muscle health data to the past literature on polyamine decline with age and muscle decline with age (although most of the literature fails to establish a causal relationship, which could be important to state- even though supplementation in rodents does seem to work...)

Round 2

Reviewer 1 Report

The authors have appropriately addressed my concerns in the revised manuscript. I especially appreciate the time taken to clarify the "age-related" changes in polyamine levels - hopefully this will help to reduce a common misinterpretation. 
